# Implementation strategies and outcomes of intravenous iron use for treatment of anaemia during and after pregnancy in low- and middle-income countries: A scoping review

Mobolanle Balogun[1,2,3]*, Elizabeth Adjoa Kumah[1,4], Victoria Olawunmi Adaramoye[1,5], Ejemai Eboreime[6], Charles Ameh[1,7], Bosede Bukola Afolabi[1,3,5]

1 Department of International Public Health, Liverpool School of Tropical Medicine, Liverpool, United Kingdom, 2 Department of Community Health & Primary Care, College of Medicine of the University of Lagos, Lagos, Nigeria, 3 Centre for Clinical Trials, Research and Implementation Science, University of Lagos, Lagos, Nigeria, 4 Foundations – What Works Centre for Children and Families, London, United Kingdom, 5 Department of Obstetrics & Gynaecology, College of Medicine of the University of Lagos, Lagos, Nigeria, 6 Department of Psychiatry, Faculty of Medicine, Dalhousie University, Halifax, Nova Scotia, Canada, 7 Department Obstetrics & Gynaecology, University of Nairobi, Nairobi, Kenya

* mbalogun@cmul.edu.ng

## Abstract

There is sufficient evidence of the efficacy of Intravenous (IV) over oral iron in treating anaemia in pregnancy and postpartum. However, poor implementation can lead to little or no benefit. The objectives of this scoping review are to map and synthesise evidence related to implementation strategies and implementation outcomes of IV iron use for treating anaemia in pregnancy and the postpartum in low- and middle-income countries (LMICs). This scoping review was conducted in accordance with the Joanna Briggs Institute's methodology for conducting scoping reviews. Electronic databases were searched to identify relevant literature sources published up until June 2025. Two independent reviewers conducted screening using the Covidence software. Descriptive statistics was used to synthesise data and findings were reported narratively. Synthesis of implementation strategies was guided by the Expert Recommendations for Implementing Change compilation. Our search yielded 4,589 publications, 20 were included in the review. Ten studies used implementation strategies, mostly "assessment for readiness and identification of barriers and facilitators" (40%; 4/10) and "promotion of adaptability" (30%; 3/10). Fourteen studies mentioned the assessment of implementation outcomes; most assessed were acceptability (42.9%; 6/14) and fidelity (35.7%; 5/14). Only five studies used any theory, model, framework (TMF) or validated measures in the implementation of strategies or assessment of outcomes. In conclusion, there is limited implementation research on the use of IV iron for the treatment of anaemia in pregnancy and the postpartum in LMICs. Critically, the use of TMFs and validated measures are deficient in current

**Data availability statement:** The data extraction template used for the scoping review is available as supporting information.

**Funding:** This scoping review was funded by the Bill & Melinda Gates Foundation; INV-032486 granted to BBA. The funders had no role in study design, data collection and analysis, decision to publish, or preparation of the manuscript.

**Competing interests:** I have read the journal's policy and the authors of this manuscript have the following competing interests: MB and BBA are co-authors in some of the papers included in the scoping review.

sources of evidence. More rigorous assessments of the implementation of IV iron for obstetric anaemia in LMICs are required to guide practice, policy and uptake.

## Introduction

Anaemia in pregnancy, defined as haemoglobin concentration less than 11 g/dL, affects two billion women globally, with a prevalence of 43% in low- and middle-income countries (LMICs) [1]. The prevalence of postpartum anaemia ranges from 50–80% in LMICs [2]. Anaemia in pregnancy has adverse consequences for mothers and neonates [3]. It can result in extreme fatigue, reduced physical and mental function, depression, increased risk of postpartum haemorrhage, and maternal death [3–7]. It also increases the risks for intrauterine growth retardation, premature delivery, and low birth weight in newborns [4,5]. Postpartum anaemia increases the risk of infection, and can lead to poor wound healing, depression in the mother, insufficient breast milk production, and reduced mother-infant bonding because of weakness and fatigue [8–11].

Iron deficiency anaemia is the most common cause of anaemia in pregnancy, accounting for 40% of anaemia cases worldwide [12]. Postpartum anaemia is caused mostly by untreated iron deficiency anaemia in pregnancy and excessive blood loss during or after birth [13]. Currently, oral iron is the standard treatment for obstetric anaemia. While oral iron supplementation has been the standard treatment for decades [14], up to 70% of women experience side effects that lead them to stop treatment [15,16]. Other barriers to oral iron supplementation include misconceptions, forgetfulness, pill burden, gender disparities, and challenges with the supply chain [17]. Intravenous (IV) iron offers a promising solution - faster correction of anaemia with fewer doses thus requiring fewer patient-provider interactions. This is advantageous in LMICs where postpartum care is limited, and where financial, material, and human resource barriers hamper the implementation of maternal health interventions [18,19]. Yet, its implementation in LMICs lags significantly behind high-income countries, where IV iron is used widely due to its accessibility, preference and acceptability, and inclusion in treatment guidelines [20–22].

This implementation gap is striking despite strong evidence supporting IV iron's efficacy, health systems in LMICs struggle to integrate it into routine care amid concerns about cost, accessibility and poorly funded health-care infrastructure and systems [23]. A critical understanding of why this implementation gap exists and how to bridge it is crucial. While clinical trials have demonstrated IV iron's effectiveness [24,25], they tell us little about the practical challenges of implementing it in resource-constrained settings. Implementation science offers a lens to examine this challenge [26]. By studying the strategies that work (or do not) in getting evidence into practice, we can identify practical solutions. A recent mixed-methods systematic review identified some barriers to and facilitators of IV iron implementation in pregnancy [27]. However, no previous review has systematically mapped how LMICs are implementing IV iron for maternal anaemia. Which implementation strategies are being used? What outcomes are being measured? These questions remain unanswered.

Implementation strategies are methods or techniques used to enhance the adoption, implementation, and sustainability of a clinical program or practice [28]. Experts in implementation science compiled and validated 73 implementation strategies organised into nine categories in the Expert Recommendations for Implementing Change (ERIC) study [28,29]. The nine categories include: a) engage consumers; b) use evaluative and iterative strategies; c) change infrastructure; d) adapt and tailor to context; e) develop stakeholder interrelationships; f) utilise financial strategies; g) support clinicians; h) provide interactive assistance; and i) train and educate stakeholders [29]. Implementation outcomes are indicators of success that are the effects of deliberate and purposive actions to implement new treatments, practices, and services. According to Proctor's conceptual framework, these outcomes include acceptability, adoption, appropriateness, feasibility, fidelity, implementation costs, penetration, and sustainability [30]. Studying these strategies and outcomes would help in understanding the process of getting evidence into practice.

This scoping review represents the first comprehensive effort to map implementation strategies and outcomes for IV iron in maternal care across LMICs. This would be an important first step in informing the objectives of future systematic reviews on the topic [31]. Understanding how to effectively implement IV iron therapy could transform the treatment of maternal anaemia in resource-limited settings, potentially saving thousands of lives and improving countless more.

The objectives of this scoping review are to map and synthesise available evidence on implementation strategies and implementation outcomes of IV iron use in the treatment of anaemia in pregnancy and the postpartum period in LMICs. Three research questions guided the scoping review; 1) What implementation strategies have been used in the administration of IV iron for treating anaemia during and after pregnancy in LMIC settings? 2) What implementation outcomes have been assessed in the use of IV iron for the treatment of anaemia during and after pregnancy? 3) How do these implementation strategies and outcomes vary according to the study population (pregnant and postpartum women), setting, and study design?

## Methods

This scoping review was conducted in accordance with the Joanna Briggs Institute (JBI) methodology for conducting scoping reviews [32]. The "Preferred Reporting Items for Systematic Reviews and Meta-Analyses extension for Scoping Review" (PRISMA-ScR) checklist guided the documentation of the screening process (S1 Checklist) [33]. A protocol for this scoping review was developed a priori and registered in Open Science Framework (https://doi.org/10.17605/OSF.IO/JWBDZ) [34]. The protocol extensively outlined the plan for the scoping review and any deviation was noted and duly reported.

### Eligibility criteria

**Participants.** We included studies that described implementation strategies and/or outcomes in the use of IV iron among pregnant and/or postpartum women of any age or ethnicity, residing in an LMIC.

**Concept.** This scoping review included studies that use IV iron of any brand, either alone or in comparison with other forms of treatment of anaemia in pregnancy and/or postpartum. The studies proposed or described implementation strategies and/or implementation outcomes in the use of the intervention.

**Context.** We included studies that were conducted in LMICs, defined as countries with a cumulative annual gross national income per capita of less than or equal to $13205 [35]. The studies included those conducted in health facilities, the community or other settings. No date restrictions were applied to the search. We excluded studies conducted in high-income countries.

### Types of studies

This scoping review considered both experimental and quasi-experimental study designs including RCTs, non-randomised controlled trials, before and after studies and interrupted time-series studies. In addition, analytical observational studies including prospective and retrospective cohort studies, case-control studies and analytical cross-sectional studies were

considered for inclusion. This review also considered descriptive observational study designs including case series, individual case reports and descriptive cross-sectional studies using quantitative or qualitative methods for inclusion. Conference abstracts containing data on IV iron implementation strategies and outcomes were also considered for inclusion in this scoping review.

**Exclusion criteria.** We excluded the following studies:

- Studies that report the administration of IV iron in general without specific administration among pregnant or postpartum women

- Studies that report the treatment of obstetric anaemia without specific information on the administration of IV iron

- Systematic, scoping or narrative reviews

### Search strategy

The databases searched were MEDLINE, CINAHL, Embase, PubMed, Global Health, Cochrane Collaboration, ScienceDirect, Scopus, and Web of Science. The sources of grey literature that were searched included the WHO International Clinical Trials Registry Platform, WHO Global Index Medicus, ClinicalTrials.gov, Google Scholar and the Bielefeld Academic Search Engine. The search was initially conducted between 30th June and 30th July, 2023 and updated between 30th May and 4th June, 2025, before completion of the review.

The search strategy was developed in consultation with an experienced medical librarian, which aimed to locate both published and unpublished studies. A combination of keywords from the research questions was used to develop a full search strategy for electronic databases.

Considering that many studies may not use implementation science terminologies since they may not be implementation science studies, our search string was only a combination of intravenous iron, pregnant women, postpartum, anaemia, low-/middle-income countries and the synonyms of these terms identified from the initial search in MEDLINE (S1 Table).

The search strategy was adapted for each included database and/or information source. The reference lists of all included sources of evidence were also screened for additional studies. Studies published in English or any other language for which a translator was available were included. Two relevant studies not published in English were translated and included; one was in Afrikaans and the other was in Spanish.

### Management of references

The search results from all the literature sources were first imported into an EndNote library (version 20) for reference management and deduplication. Thereafter, the search results were exported into Covidence software (a web-based platform for streamlining the production of systematic/scoping reviews) for screening [36]. Covidence software also helped to remove duplicate articles that were not identified in Endnote. A few duplicates that were not identified by Covidence and Endnote software were removed manually.

### Study/source of evidence selection

Two independent reviewers (MB and VA) screened the titles and abstracts. Subsequently, studies that remained eligible after this phase underwent full text screening against the inclusion criteria by two independent reviewers (MB and EE). The reasons for exclusion of studies at the full-text stage were recorded for each excluded study. Specifically, at the full-text screening stage, studies where an implementation strategy was not used to facilitate the use of IV iron among pregnant and/or postpartum women in an LMIC, and/or that did not assess implementation outcomes of IV iron use among pregnant and/or postpartum women in LMICs were excluded. Any disagreements that arose between the reviewers at

each stage of the selection process were resolved through discussion. The results of the search and the literature screening process were presented in the PRISMA flow diagram [37].

## Data extraction

Data were extracted from the included studies by one reviewer (MB) and verified by another reviewer (EE) using a data extraction form developed in Microsoft Excel by the reviewers. The form was used primarily to collect the following information: first author, year of publication, title of study, aim of study, country, study type, study design, setting, type of participants, gestational age/trimester/number of days or weeks postpartum, grade of anaemia, type of IV iron, implementation strategies described, ERIC classification of strategies, targets of strategies, key findings of strategies, implementation outcomes described, targets of outcomes, methods of data collection of outcomes, and key findings of outcomes. The full data extraction template is included as an additional file (S1 Data).

The developed data extraction form was pretested on 20% of the included studies and was modified as necessary before commencing data extraction. Any disagreements that arose between the reviewers were resolved through discussion. The final results were reviewed by experts on the review team (EAK, CA and BBA).

## Data synthesis

The extracted data were exported from Excel into the Stata software (version 15.1) for synthesis. The implementation strategies identified were classified on the basis of the ERIC compilation and further synthesised into the nine ERIC categories. The implementation outcomes were reported as identified in the included studies except for compliance, which was reclassified as 'fidelity' [38]. Additionally, patient satisfaction with therapy and reasons for use and non-use were reclassified as 'acceptability' [39]. Cost was broadened to include implementation costs (cost of executing implementation strategies and interventions costs (costs directly associated with the use of IV iron), which both have economic consequences for implementing IV iron [40].

Descriptive statistics were used to synthesise and report the characteristics and outcome measures from the included studies. These are defined below:

**Characteristics of the included studies.**

- Country of study: Name of country

- Year of publication: This was categorized into three groups; 1) studies published before 2010; 2) studies published between 2010 and 2020; and 3) studies published after 2020.

- Type of evidence source: Original research, study protocol, or trial registration.

- Study design: This was grouped into experimental studies (RCTs and quasi-experimental studies) and observational studies (comparative study, implementation study, quality improvement study, formative qualitative study, prospective study, cross-sectional study, retrospective analysis of routine data, cost-effective analysis, and registry)

- Study setting: health facility (hospitals or clinics), community and online.

- Type of IV iron in study: Name of IV iron

- Target population for IV iron: Pregnant or postpartum women

- Duration of pregnancy or postpartum: Pregnancy gestational age in weeks was categorised into trimesters. Postpartum period was documented in hours, days or weeks.

- Grade of anaemia: This was classified as; 10–10.9 g/dl (mild anaemia), 7–9.9 g/dl (moderate anaemia), and <7 g/dl (severe anaemia) [41].

**Global Public Health**

**Outcome measures.**

- Type of implementation strategy: Classified using ERIC compilation and synthesised under the first seven categories. The last two categories, "support clinicians" and "utilise financial strategies," were omitted during data synthesis, as there were no relevant implementation strategies described that fit both.

- Type of implementation outcome: Proctor's outcomes [30].

Findings were reported narratively using tables and figures. As this is a scoping review, there was no quality assessment of the included studies [42].

## Results

The results of the search and the screening process are presented in Fig 1. Our search yielded 4,589 references. A total of 4,447 articles were obtained from nine databases, and 142 references were obtained from four grey literature sources. After removal of duplicates, 3,601 studies were included in the title and abstract screening. Of these, 188 studies were sought for retrieval but 16 of the full texts were unavailable, leaving 172 studies that were assessed for eligibility. At the end of the full-text screening, 150 studies were excluded for different reasons (Fig 1), while 22 studies were eligible. Two of these eligible studies were ongoing without any formative results presented and were excluded [43,44], leaving 20 studies for the final review (45–64) Three of these studies assessed different outcomes within the same project [45–47], while another two studies assessed different outcomes within another project [55,56].

## Characteristics of the included studies

The 20 included studies were from Asia or Africa and were distributed as follows: 13 (65%) from India [48–54,58–62,64], 3 (15%) from Nigeria [45–47], 3 (15%) from Malawi [55–57], and 1 (5%) from Malaysia [63]. Half (50%; 10/20) of the studies were published after 2020, almost all (95%; 19/20) were original research while one was a study protocol that had formative results [45]. There were a variety of study designs, the most common being prospective studies (20%; 4/20) [52,59,60,62]. The majority (90%; 18/20) of the studies were conducted in health facility settings and the type of IV iron used was mostly iron sucrose (60%; 12/20) [Table 1].

The target population was pregnant women in 16 studies (80%) and a combination of pregnant and postpartum women in four studies (20%). Four studies described the use of IV iron in the second trimester [48,57,59,64], seven in the second to third trimesters [45–47,51,53,54,61], four in the third trimester [55,56,58,63], and one in all trimesters [49], whereas three did not report the timing of giving IV iron during pregnancy [50,52,60]. None of the four studies that included postpartum women reported the time of IV iron administration during the postpartum period. The grade of anaemia among pregnant or postpartum women was moderate to severe in twelve studies [45–47,51,53,55–57,59–62] studies, moderate in seven studies [48–50,54,58,63,64], and severe in one study [52].

## Outcome measures

**Types of implementation strategies.** Ten studies (eight observational [47–49,51,54–57] and two experimental [45,53] mentioned the use of implementation strategies in the administration of IV iron. The most common strategies were "assessment for readiness and identification of barriers and facilitators" (40%; 4/10) [45,55–57] and "promotion of adaptability" (30%; 3/10) [51,53,54]. The strategies used in the experimental studies were from the categories "use evaluative and iterative strategies", "adapt and tailor to context", and "develop stakeholder interrelationships". The strategies used in observational studies fall into the categories "use evaluative and iterative strategies", "provide interactive assistance", "adapt and tailor to context", "develop stakeholder interrelationships", "train and educate stakeholders", "engage consumers" and "change infrastructure" (Table 2). The targets of the strategies were patients

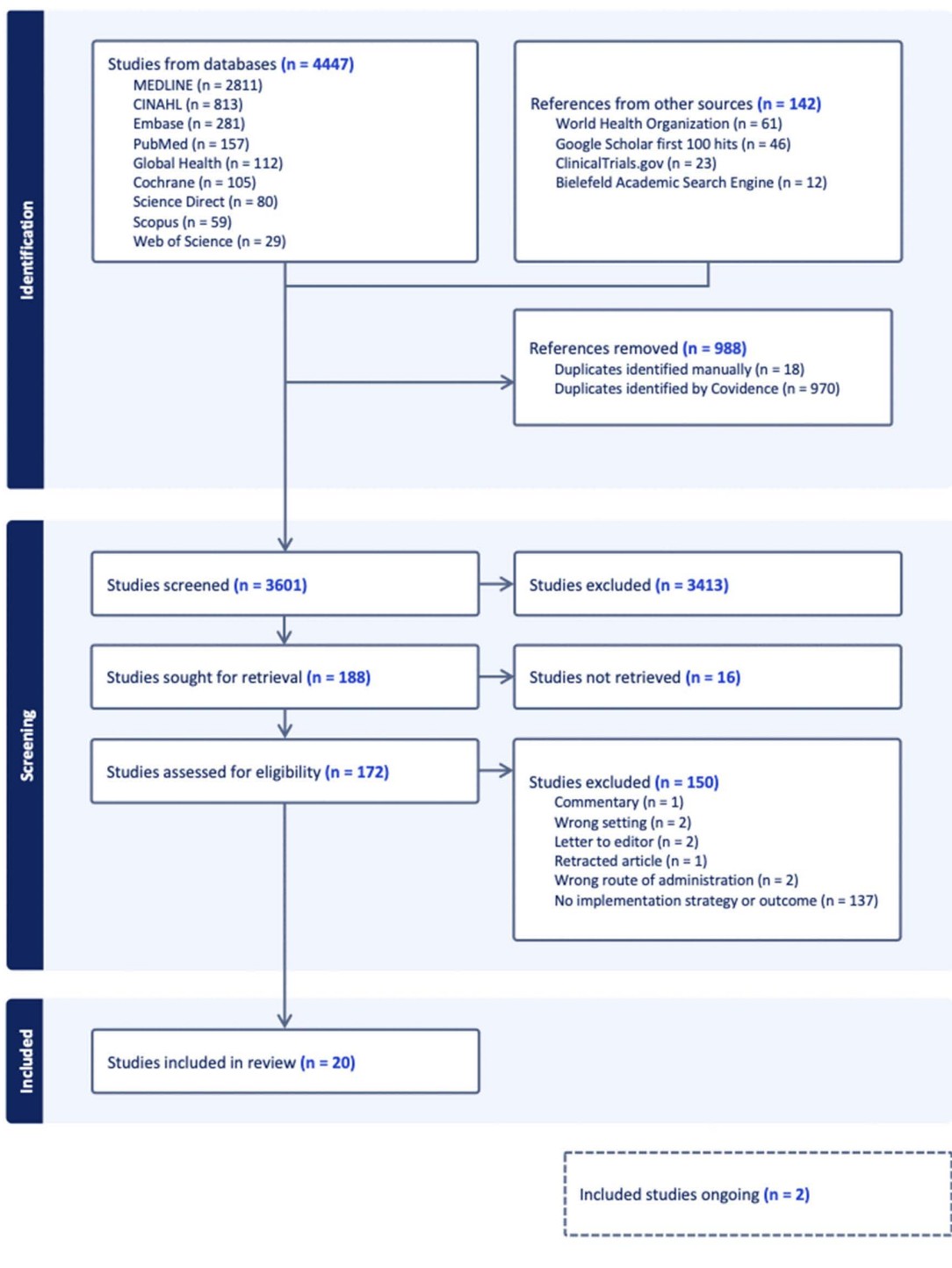

**Fig 1. PRISMA flow chart of study selection.**

**Table 1. Characteristics of the included studies.**

| Classification | Frequency (percent) n = 20 |
|---|---|
| *Country of study* | |
| India | 13 (65) |
| Nigeria | 3 (15) |
| Malawi | 3 (15) |
| Malaysia | 1 (5) |
| *Year of publication* | |
| Before 2010 | 1 (5) |
| Between 2010 and 2020 | 9 (45) |
| After 2020 | 10 (50) |
| *Type of evidence source* | |
| Original research | 19 (95) |
| Study protocol | 1 (5) |
| *Study design* | |
| Prospective study | 4 (25) |
| Randomised control trial | 3 (15) |
| Qualitative study | 3 (15) |
| Cross-sectional study | 3 (15) |
| Retrospective analysis of the routine clinical data | 2 (10) |
| Comparative study | 1 (5) |
| Mixed-method study | 1 (5) |
| Cost-effectiveness analysis in an RCT | 1 (5) |
| Registry | 1 (5) |
| Experienced-based co-design approach | 1 (5) |
| *Study setting* | |
| Health facility | 18 (90) |
| Community | 1 (5) |
| Online survey | 1 (5) |
| *Type of IV iron* | |
| Ferric carboxymaltose | 6 (30) |
| Iron sucrose | 12 (60) |
| Ferric carboxymaltose and iron sucrose | 1 (5) |
| Ferric carboxymaltose and Ferric derisomaltose | 1 (5) |

RCT: Randomised controlled trial.

(80%; 8/10) [45,48,51,53–57], healthcare providers (50%; 5/10) [45,47,51,55,57], family (30%; 3/10) [45,55,56], policymakers (20%; 2/10) [45,55], community (10%; 1/10) [55], and health system (10%; 1/10) [49].

Table 3 documents the key findings of implementation strategies where reported. The barriers to IV iron implementation from the demand-side (patient and community) were centred around misconceptions, local attitudes, lack of community sensitization, long distance to health facilities, non-compliance to appointments, lack of male involvement and support, cost of IV iron and discomfort associated with the treatment [45,55–57] Barriers from the supply-side (provider and health system) were inadequate knowledge, lack of capability to deliver iron, and long waiting times [55,56]. An experimental study identified clarity of information, periodic provider training and involvement at all social levels as facilitators for IV iron treatment during the trial [45]. Single-dose treatment and the perception of improved health were identified as facilitators

**Table 2. Implementation strategies used in the included studies stratified by study design.**

| Types of implementation strategies using ERIC compilation | All studies n=10 Freq (%) | Observational studies n=8 Freq (%) | Experimental studies n=2 Freq (%) |
|---|---|---|---|
| **Use evaluative and iterative strategies** | | | |
| Assess for readiness and identify barriers and facilitators [45,55–57] | 4 (40) | 3 (37.5) | 1 (50) |
| **Provide interactive assistance** | | | |
| Provide clinical supervision [47,51] | 2 (20) | 2 (25) | |
| **Adapt and tailor to context** | | | |
| Tailor strategies [45] | 1 (10) | | 1 (50) |
| Promote adaptability [51,53,54] | 3 (30) | 2 (25) | 1 (50) |
| **Develop stakeholder interrelationships** | | | |
| Conduct local consensus discussions [55] | 1 (10) | 1 (12.5) | |
| Use advisory boards and workgroups [45] | 1 (10) | | 1 (50) |
| **Train and educate stakeholders** | | | |
| Conduct ongoing training [47] | 1 (10) | 1 (12.5) | |
| Distribute educational materials [47] | 1 (10) | 1 (12.5) | |
| **Engage consumers** | | | |
| Involve patients and family members [55] | 1 (10) | 1 (12.5) | |
| **Change infrastructure** | | | |
| Change record system [49] | 1 (10) | 1 (12.5) | |
| Change physical structure and equipment [47] | 1 (10) | 1 (12.5) | |

in an observational study [57]. Adapting treatment guidelines to give pregnant women with severe anaemia IV iron when they refuse blood transfusion increased uptake of IV iron and reduced severe anaemia [51,54]. Local consensus discussions prior to commencement of a trial generated strategies to mitigate identified barriers [55].

Four studies used a theory, model or framework (TMF) in the implementation of strategies, namely, the Consolidated Framework for Implementation Research (CFIR) [45,55], Patient-Centered Access to Healthcare (PCAH) [56], and the ERIC tool [55] (Table 3).

**Types of implementation outcomes.** Fourteen studies assessed implementation outcomes [45–47,49,50,52,57–64]. Thirteen of these were observational studies (92.9%). The most common implementation outcomes assessed were acceptability (42.9%; 6/14) [45,46,50,57,60,64] and fidelity (35.7%; 5/14) [47,49,52,58,63]. The one experimental study assessed acceptability and feasibility at the formative phase of the trial [45]. The observational studies assessed all varieties of implementation outcomes found in the review, mostly acceptability (38.5%; 5/13) and fidelity (38.5%; 5/13) [Table 4]. The targets of the implementation outcomes were patients (78.6%; 11/14) [45,46,49,52,57,58,60–64], healthcare providers (42.9%; 6/14) [45–47,49,50,57], health system (21.4%; 3/14) [59,61,62], family (14.3%; 2/14) [45,46], and policymakers ((14.3%; 2/14) [45,61].

The data collection methods and key findings of the assessments of implementation outcomes are documented in Table 5. Five studies that reported findings from assessing acceptability showed that IV iron was acceptable to both patients and providers [46,50,57,60,64]. Studies that reported findings from assessing fidelity to multiple-dose iron sucrose revealed gaps in compliance with the full course of treatment among pregnant and postpartum women [49,52,58,60]. One study that assessed fidelity to single-dose ferric carboxymaltose among skilled health personnel found moderate adherence to protocol [47]. Providers in one study reported that IV iron is feasible [52]. Assessment of cost revealed that IV iron is costlier but more effective than oral iron [61,62]. However, it is less expensive to administer as a bolus than as an infusion [59].

**Table 3. Key findings of implementation strategies.**

| Types of implementation strategies | Theories, Models or Frameworks | Key findings |
|---|---|---|
| Assess for readiness and identify barriers and facilitators [45,55–57] | CFIR [45] | • Barrier: Threat of misinformation and conspiracy theories<br>• Facilitators: provision of clear, standard reference information for participants and family members; periodic training for providers; involvement of all social levels |
| | CFIR [55] | • Barriers: cost of IV iron, the lack of available resources and knowledge, local attitudes including myths and misconceptions and keeping pregnancy a secret, local conditions, lack of political will, the lack of capability of providers to deliver IV iron, and the lack of male involvement to support pregnant women's access to antenatal care. |
| | Patient-Centered Access to Healthcare [56] | • Supply-side barriers were lack of transparency in trial procedure, lack of continued community sensitisation about IV iron, long distances from home to the health facilities, long waiting times, and pregnant women non-compliance with appointments.<br>• Demand-side barriers were myths and misconceptions, cultural norms, lack of social and financial support from husbands, and physical discomfort when receiving IV iron.<br>• Supply-side facilitators were availability of clear information, pregnant women not pressured to participate, flexible opening hours and appointments, perceived effectiveness and benefits of IV iron and providers' interpersonal skills.<br>• Demand-side facilitators were health literacy about anaemia, social value and a sense of autonomy, peer support, social and financial support from family or husband, and caregiver support. |
| | None [57] | • Barriers: Superstitious narratives and perceptions about the research and drawing blood; confusion over the colour of the treatment; cost associated with procuring the treatment and how the treatment is administered, e.g., pain, fear of injection and time to administer IV iron.<br>• Facilitators: single treatment dose and perceived improvements in health and maternal-child health outcomes. |
| Promote adaptability [49,52] | None [51] | • 92 women with severe anaemia received IV iron as a result of adapting treatment guideline.<br>• The number of pregnant women with severe anaemia declined from 92 to 13 at endline (85.9% reduction) |
| | None [54] | • 14 women with severe anaemia received IV iron as a result of adapting treatment guideline. |
| Conduct local consensus discussions [55] | ERIC tool | • The proposed strategies to mitigate the barriers for the successful implementation included providing financial strategy, developing stakeholder relationships, training and educating stakeholders, supporting clinicians, and engaging end-users. |
| Distribute educational materials [47] | None | • Readily available charts and protocols contributed to high performance in highly adherent facilities. |
| Change physical structure and equipment [47] | None | • Resuscitation materials were not used as intended at several sites largely because of non-proximity to point of ferric carboxymaltose administration |

CFIR: Consolidated Framework for Implementation Research; IV: Intravenous; ERIC: Expert Recommendations for Implementing Change.

**Table 4. Implementation outcomes assessed in the included studies stratified by study design.**

| Types of implementation outcomes | All studies n = 14 Freq (%) | Observational studies n = 13 Freq (%) | Experimental studies n = 1 Freq (%) |
|---|---|---|---|
| Acceptability [45,46,50,57,60,64] | 6 (42.9) | 5 (38.5) | 1 (100) |
| Fidelity [47,49,52,58,63] | 5 (35.7) | 5 (38.5) | |
| Cost [59,61,62] | 3 (21.4) | 3 (23.1) | |
| Feasibility [45,57] | 2 (14.3) | 2 (15.4) | |
| Adoption [50] | 1 (7.1) | 1 (7.7) | |

Only three studies used any TMF or validated measures in the assessment of outcomes [45–47]. The frameworks used in these studies were CFIR, consolidated framework for implementation fidelity and theoretical framework of acceptability. The validated measures were Acceptability of Intervention Measure and Feasibility of Intervention Measure (Table 5).

## Discussion

The use of IV iron in the treatment of anaemia in pregnancy and postpartum could reduce maternal morbidity and mortality in LMICs. Our scoping review sought to map out the implementation strategies and outcomes used in the administration of IV iron in LMICs. The most common implementation strategies were identification of barriers and facilitators and promotion of adaptability, whereas the most common implementation outcomes were acceptability and fidelity. A higher number of observational than experimental studies used implementation strategies in the administration of IV iron, and almost all the studies that assessed implementation outcomes were observational studies.

Our scoping review also revealed that most of the studies that assessed IV iron implementation were from India. India has been at the forefront in conducting several studies exploring the safety and efficacy of intravenous and intramuscular iron since the 1970s, possibly due to its high prevalence of anaemia in pregnancy [65,66].

The field of implementation science was formalised in 2006 with the emergence of a new journal aimed at bridging the implementation gap between research evidence and adoption into practice [67]. The field has since gained ground with established methods and frameworks. Thus, it is not surprising that most of the studies in this review were conducted after 2010 (half of these after 2020) and only one before 2010. This coincides with the period of rapid increase in implementation research as well as research on IV iron during pregnancy globally, with an appreciable increase in LMICs [27,68]. This reflects the availability of newer IV iron formulations with fewer side effects than older, high-molecular-weight formulations [69].

A variety of observational study designs were used in this review. These are easier and less costly to conduct than RCTs, address important research questions not suitable for RCTs and provide preliminary data to justify an RCT [70]. One of the three RCTs included in this review collected formative data on implementation outcomes in a hybrid type 1 effectiveness-implementation design [45]. This blended design provides benefits over research that looks at effectiveness and implementation independently [71]. As more RCTs on the topic are conducted in LMICs, hybrid designs are recommended to facilitate the implementation of IV iron in the treatment of obstetric anaemia. The general lack of implementation strategy or implementation outcomes assessment in experimental studies designs in this review could result in delays to bridging critical implementation gaps in LMICs, particularly as they incur heavier resource investments than observational studies [72].

Identification of barriers and facilitators as an implementation strategy helps in understanding contextual impediments and strengths to implementation efforts [28]. In LMICs, the practicability of this strategy could be hindered by scarce resources and expertise [19]. However, the barriers to the implementation of maternal health evidence are similar in LMICs [19]. Therefore, lessons learned from a subset of LMICs may be applicable to others. The barriers identified in our

**Table 5. Data collection methods and key findings of the assessment of implementation outcomes.**

| Types of implementation outcomes | Data collection methods | Theories, Models, Frameworks or validated measures | Key findings |
|---|---|---|---|
| Acceptability [45,46,50,57,60,64] | Surveys, FGDs, KIIs (N = 193) [45] | Acceptability of Intervention Measure | Not reported |
| | FGDs, KIIs (N = 169) [46] | CFIR and theoretical framework of acceptability | • Generally, all stakeholders had a positive affective attitude towards IV iron based on its comparative advantages to oral iron. |
| | Online survey (N = 107) [50] | None | • The main reasons providers used IV iron sucrose were efficacy (83, 77.57%), less blood transfusions needed to be done (79, 73.83%), no adverse events (58, 54.21%).<br>• The reason for non-use was cost (69, 64.49%). |
| | FGDs, IDIs (N = 36) [57] | None | • Despite perceived concerns and challenges in FCM implementation, pregnant women and providers' reflections suggest IV iron is acceptable. |
| | Likert scale assessment of patient's satisfaction (1- Excellent, 2- Good, 3- Average, 4- Poor and 5- Very poor) (N = 60) [60] | None | • Group A (high dose IV iron sucrose) pregnant women had excellent satisfaction among 53%, good in 20%, average in 6.6% and poor in 6.6% of patients.<br>• Group B (multiple dose IV iron sucrose) had excellent satisfaction in 40%, good in 27%, average in 20% and poor in 13% of patients. |
| | Interview to assess "like" and "dislike" of therapy (N = 100) [64] | None | • Acceptability for IV therapy was higher than oral therapy though not statistically significant.<br>• 78% of pregnant women on oral iron liked the therapy<br>• 86% on IV iron sucrose liked the therapy. |
| Fidelity [47,49,52,58,63] | Observation of FCM administration (N = 254); IDIs (N = 14) [47] | Consolidated framework for implementation fidelity | • Adherence to FCM administration as per protocol was moderate (63%) and varied depending on facility level.<br>• The lowest level of adherence was observed in PHCs (36%). |
| | Registry (N = 764) [49] | None | • In Tamil Nadu (TN), 387(61.8%) pregnant and postpartum women received two doses of iron sucrose<br>• In Uttar Pradesh (UP), 45(32.6%) patients received two doses.<br>• The median duration of infusion was 30 min [25–40] in both states.<br>• The median gap between 2 doses was 3 days in TN and 2 days in UP.* |
| | Observation of follow-up and phone call enquiry (N = 406) [52] | None | • Most multigravida (74.2%) and primigravida (65.3%) pregnant women interrupted iron sucrose treatment before 6 complete doses. |
| | Interview with structured questionnaire; review of records with pregnant women (N = 350) [58] | None | • 276 (79%) pregnant women complied with IV iron sucrose treatment.<br>• Of these, 31 (8.9%) received six doses.<br>• Among the 74 (21%) women who were noncompliant, 39 (11.1%) did not receive even one dose.* |
| | Facility record (N = 120) [63] | None | • 10 patients (8.3%) did not complete their therapy (eight antenatal mothers delivered before completion of treatment; two postpartum patients defaulted their follow-up). |
| Feasibility [45,57] | Surveys, FGDs, KIIs (N = 193) [45] | Feasibility of Intervention Measure | Not reported |

*(Continued)*

**Table 5.** (Continued)

| Types of implementation outcomes | Data collection methods | Theories, Models, Frameworks or validated measures | Key findings |
|---|---|---|---|
| | IDIs (N = 36) [57] | None | • Providers saw the implementation of IV iron in pregnancy as highly feasible.<br>• However, there were considerable concerns about health system issues that would need to be addressed before full-scale. |
| Cost [59,61,62] | Not reported (N = 500) [59] | None | • Multiple doses of IV iron sucrose given as bolus was much cheaper than multiple doses of IV iron sucrose infusion to treat pregnant and postpartum women. |
| | User costs for accessing treatment, hospital costs for providing care, communication with Ministry [61] | None | • IV iron sucrose was found to be more costly but more effective than the oral iron folic acid tablet for treatment of severe anaemia. |
| | User costs from financial records, field interviews. Cost of therapy from government-rate contract, local bulk procurement. Consumables cost from facility. Travel and wage losses from field records [62] | None | • IV iron sucrose therapy was more cost-effective than oral iron therapy among pregnant women for management of moderate and severe anaemia. |
| Adoption [50] | Online survey (N = 107) | None | • 62 providers (57.94%) used IV iron sucrose for severe anaemia when the patient showed intolerance to oral iron. |

FGD: Focus group discussion; KII: Key informant interview; IDI: Indepth interview; IV: Intravenous; FCM: ferric carboxymaltose.

\* Standard treatment is IV infusion of 100 mg iron sucrose in 100 ml of normal saline for 20–30 min once a day for four days with 2–4 days of interval between each infusion within a period of two weeks. A moderately anaemic pregnant woman who has completed four of iron sucrose injections as per guidelines is said to be compliant to treatment [58].

review were related to both supply and demand, and future implementation of IV iron should address these. Formative assessment in an experimental study identified the need for clear information and involvement at all social levels [45], which could address several barriers reported in our review.

Promotion of adaptability, which identifies ways an intervention can be tailored to meet local needs [28], was used in three studies in this review. Two studies adapted IV iron treatment during pregnancy to treat severely anaemic women who rejected blood transfusion [51,54]. The third study increased the IV iron infusion time from 15 to 30 minutes due to limited safety data [53]. Interventions can be adapted throughout the implementation process due to diverse contexts and needs, particularly in LMICs with weak health-care systems and lack of sustained political will [23,73]. There is now a better understanding of adaptation in the context of implementation with accompanying frameworks [74], which would be useful in future documentation of adaptation of IV iron use in LMICs.

Implementation strategies from seven out of the nine ERIC categories were represented in our review, whereas the categories labelled "support clinicians" and "utilise financial strategies" were not represented. Using financial strategies would be critical if IV iron is to be used sustainably in LMICs. Providers and patients in Nigeria have expressed concerns about out-of-pocket costs of IV iron particularly among poor women [46]. Only one observational study reported ongoing training of skilled health personnel as a strategy [47]. There are training deficiencies in LMICs and educational strategies can address misconceptions and propagate the safety profile awareness of new IV iron formulations [27]. While conducting studies on the topic, consumer engagement is essential throughout the research continuum, rather than just the execution phase of the research as seen in some RCTs [75]. Evidence from a systematic review suggests that this is beneficial, particularly consumer involvement in developing patient information materials [76].

None of the studies referred to quality monitoring systems in the administration of IV iron to pregnant and postpartum women. Historically, the quantity rather than the quality of health services has been the focus in developing countries, although quality is crucial for improved health outcomes [77]. As the use of IV iron for obstetric anaemia in LMICs increases, strengthening of quality improvement and monitoring systems is recommended. The targets of the implementation strategies were mainly patients and providers. Although, implementation strategies targeted at healthcare providers can lead to improvements in obstetric care in LMICs [78], a multi-pronged approach across the socioecological levels at the system, community, organisation, individual, and policy levels may be beneficial [79].

Generally, most of the studies measured acceptability as an implementation outcome and found IV iron to be acceptable despite identified barriers. Understanding IV iron acceptability from the perspectives of providers and recipients is key in the design, evaluation and implementation of interventions, and can help pave the way for its uptake and routine use in LMIC settings [39,46]. With respect to fidelity, most of the studies that assessed it looked at compliance of patients with multidose IV iron treatment [49,52,58,63]. Fidelity, which is the degree to which an intervention is implemented as intended, provides a viable assessment of the contribution of an intervention to outcomes or if outcomes are due to unknown or external factors [38,80]. Fidelity assessment is in its infancy and evidence suggests that it is not a consistent practice in cluster randomised trials of public health interventions in LMICs [81]. To ensure the quality use of IV iron in obstetric anaemia across different levels of the health system in LMICs, fidelity assessment should be a standard component in future research.

Five out of eight outcomes from Proctor's original taxonomy were represented in this review. However, only a few implementation outcomes were used consistently across the studies, and data on long-term outcomes, such as maintenance and sustainability, were lacking. Future studies should build on this by incorporating more relevant outcomes beyond acceptability, fidelity, feasibility, cost and adoption.

Like implementation strategies, implementation outcomes were measured mainly among patients and providers. As implementation outcome research advances in LMICs, consideration of other stakeholders in the socioecological model is important.

A significant limitation of studies on IV iron administration for obstetric anaemia in LMICs is the dearth of TMFs or validated measures for implementing strategies or assessing outcomes. This is possibly because most of the studies were not implementation science-oriented yet assessed outcomes such as acceptability, fidelity, feasibility and cost. Over time, implementation science has progressed from being empirically driven to the use of TMFs to gain insights into the mechanisms by which implementation is likely to succeed [82]. The use of TMFs promotes shared understanding and guides implementation planning and the choice of strategies, and there are an increasing number of options [83]. Despite the limitations found in their usability, application and testability [83], we recommend the use of TMFs and other validated measures to guide implementation planning of IV iron intervention for obstetric anaemia in LMICs.

Another gap in studies on IV iron administration for obstetric anaemia in LMICs is the limited implementation assessment of IV iron in the postpartum period. This fragile period is characterised by high burden of anaemia, high attrition from postnatal care services, and inadequate follow-up care by limited numbers of providers in LMICs [2,84]. The context of IV iron implementation likely differs in the antenatal and postnatal periods, thus research and policies that emphasise IV iron implementation in the postpartum women are critical.

## Strengths and limitations of the review

To the best of our knowledge, this is the first scoping review on this topic with an extensive search of multiple databases and grey literature. We also translated the two studies that were published in Afrikaans and Spanish. However, being a scoping review, quality assessment of the included studies was not performed; thus, gaps related to the low quality of research could not be identified.

We set out to map how implementation strategies and outcomes vary according to study population, setting, and study design. However, little or no variation was found in the study population and settings, as most of the studies were among pregnant women as opposed to postpartum women and in health facility settings. Thus, that research question could not be fully explored.

## Conclusions

This review has shown that despite the growing number of studies on IV iron use treating anaemia during and after pregnancy in LMICs, there are few sources of evidence on the use of implementation strategies and outcomes in the use of the intervention among the population. Additionally, the current evidence is from only four LMICs. The implementation strategies and outcomes described have a limited variety of target stakeholders and experimental studies are limited by their assessment of implementation outcomes. In particular, the use of TMFs and validated measures are grossly deficient in current sources of evidence. More rigorous assessments of the implementation of IV iron intervention in obstetric anaemia across more LMICs are needed to guide practice and policy. Implementation science frameworks and methods should guide future studies, which should consider long-term implementation outcomes.

## Supporting information

**S1 Checklist. PRISMA-ScR checklist.**
(DOCX)

**S1 Table. Full search strategy for MEDLINE.**
(DOCX)

**S1 Data. Data extraction template.**
(XLSX)

## Acknowledgments

The authors acknowledge Alison Derbyshire, who helped with the development of the search strategy for this scoping review.

## Author contributions

**Conceptualization:** Mobolanle Balogun, Elizabeth Adjoa Kumah, Charles Ameh, Bosede Bukola Afolabi.

**Formal analysis:** Mobolanle Balogun, Ejemai Eboreime.

**Investigation:** Mobolanle Balogun, Elizabeth Adjoa Kumah, Victoria Olawunmi Adaramoye, Ejemai Eboreime.

**Methodology:** Mobolanle Balogun, Elizabeth Adjoa Kumah, Victoria Olawunmi Adaramoye, Ejemai Eboreime, Charles Ameh, Bosede Bukola Afolabi.

**Resources:** Bosede Bukola Afolabi.

**Supervision:** Elizabeth Adjoa Kumah, Charles Ameh, Bosede Bukola Afolabi.

**Writing – original draft:** Mobolanle Balogun.

**Writing – review & editing:** Elizabeth Adjoa Kumah, Victoria Olawunmi Adaramoye, Ejemai Eboreime, Charles Ameh, Bosede Bukola Afolabi.

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
