## [Decision Letter · Decision Letter 0]

27 Aug 2025

PGPH-D-25-01513

Implementation strategies and outcomes of intravenous iron use for treatment of anaemia during and after pregnancy in LMICs: a scoping review

Dear Dr. Balogun,

Thank you for submitting your manuscript to PLOS Global Public Health. After careful consideration, we feel that it has merit but does not fully meet PLOS Global Public Health’s publication criteria as it currently stands. Therefore, we invite you to submit a revised version of the manuscript that addresses the points raised during the review process.

Reviewers Comments : 

Reviewer 1 

1. The search strategy to begin with itself is very incomplete where the important terms to check implementation strategies or outcomes were not looked for. This has resulted in a very vague identification of studies which were designed to look for clinical outcomes than implementation outcomes. How can formative study describing barriers and facilitators be considered as implementation strategies...these would provide basis for developing implementation strategies which are completely not available nor described. The authors understanding on implementation research sounds quite good.However they have missed the very essence in their search strategy.

2. There is no information about baseline situations in the studies in terms of availability of IV Iron type

3. The strategies described are inadequate to make any meaningful conclusions for eg. change in infrastructure - what change was done, at what level? whether this change was specific to providing IV Iron ?

4. No mention of the timelines needed for implementation

5. There is a mention of cost of IV iron as a major barrier...which is known. What implementation strategies were undertaken to address this?

6. This entire exercise needs to be repeated to clearly justify the review based on the objectives the authors set out to achieve

Reviewer 2

Thanks for the opportunity to review this narrative review of implementation strategies and outcomes of intravenous iron during and after pregnancy in LMICs. I believe that this review serves an important function by compiling results on this topic in one place, especially as multiple trials of FCM IV iron in pregnancy and postpartum along with accompanying implementation research are ongoing in LMICs. I would like to offer the following comments for the authors’ consideration.

Title – consider spelling out LMICs in the title

Line 22 – Removed “the” before postpartum.

Line 31 – If this is a narrative review, what was the role of descriptive statistics? Was this used for categorization or counting?

Line 55 and 59 – Fatigue is mentioned on both lines. Suggest combining or mentioning in only one place.

Line 67 – Women also stop taking oral iron for other reasons, including misconceptions, forgetfulness, challenges with getting adequate supply, and socioeconomic barriers.

Line 342 – 42.9% would not be considered most. The authors may want to consider rewording this.

Lines 360-361 – Please check if part of this sentence is missing.

Table 3 – This is a useful table. I have some suggestions about making it easier to read. In the types of implementation strategies column, the authors could consider making a heading for each category, especially those that repeat (e.g., Assess for readiness and identify barriers and facilitators), instead listing it several times. In the key findings column, consider using bullet points within each row (e.g., separate bullets for barriers and facilitators).

Table 5 – Similar to my comment on Table 3, in the types of implementation outcomes column, the authors could consider having heading for each outcome (e.g., acceptability, fidelity, feasibility, cost) instead of repeating in each row. The second to last row of the table has the name of an author in the types of implementation outcomes column. For the fidelity (46) row, it’s hard to understand the results in the key findings column because we don’t know what the expected duration of infusion or number of days between doses was. Cost (56) – IV iron sucrose as bolus compared to what (e.g., how many doses for the IV infusion)?

Line 428 – What does SHPs stand for?

Discussion – Consider shortening the discussion. It is quite long and could be trimmed without losing meaning.

Line 490 – Would the authors consider different countries as different settings? If so, then there is some variability in the setting.

Figures 2-4 – It’s nice to have some figures, but the information across these three figures could be shared more succinctly in a single table. Also, I noticed in Figure 2 that this refers to articles rather than studies. In some cases, I believe that some of the articles from a country are part of the same study. Would it make sense throughout the paper to refer to the number of studies rather than the number of articles?

We look forward to receiving your revised manuscript.

Kind regards,

Rahul Gajbhiye, MBBS PhD

Academic Editor

Journal Requirements:

i. State the initials, alongside each funding source, of each author to receive each grant.

ii. State what role the funders took in the study. If the funders had no role in your study, please state: “The funders had no role in study design, data collection and analysis, decision to publish, or preparation of the manuscript.”

2. Please send a completed 'Competing Interests' statement, including any COIs declared by your co-authors. If you have no competing interests to declare, please state "The authors have declared that no competing interests exist". Otherwise please declare all competing interests beginning with the statement "I have read the journal's policy and the authors of this manuscript have the following competing interests:"

3. Please provide separate figure files in .tif or .eps format.

4. Some material included in your submission may be copyrighted. According to PLOS’s copyright policy, authors who use figures or other material (e.g., graphics, clipart, maps) from another author or copyright holder must demonstrate or obtain permission to publish this material under the Creative Commons Attribution 4.0 International (CC BY 4.0) License used by PLOS journals. Please closely review the details of PLOS’s copyright requirements here: PLOS Licenses and Copyright. If you need to request permissions from a copyright holder, you may use PLOS's Copyright Content Permission form.

Potential Copyright Issues:

Figure 2: please (a) provide a direct link to the base layer of the map (i.e., the country or region border shape) and ensure this is also included in the figure legend; and (b) provide a link to the terms of use / license information for the base layer image or shapefile. We cannot publish proprietary or copyrighted maps (e.g. Google Maps, Mapquest) and the terms of use for your map base layer must be compatible with our CC-BY 4.0 license. 

Additional Editor Comments (if provided):

Reviewer 1

1. The search strategy to begin with itself is very incomplete where the important terms to check implementation strategies or outcomes were not looked for. This has resulted in a very vague identification of studies which were designed to look for clinical outcomes than implementation outcomes. How can formative study describing barriers and facilitators be considered as implementation strategies...these would provide basis for developing implementation strategies which are completely not available nor described. The authors understanding on implementation research sounds quite good.However they have missed the very essence in their search strategy.

2. There is no information about baseline situations in the studies in terms of availability of IV Iron type

3. The strategies described are inadequate to make any meaningful conclusions for eg. change in infrastructure - what change was done, at what level? whether this change was specific to providing IV Iron ?

4. No mention of the timelines needed for implementation

5. There is a mention of cost of IV iron as a major barrier...which is known. What implementation strategies were undertaken to address this?

6. This entire exercise needs to be repeated to clearly justify the review based on the objectives the authors set out to achieve

Reviewer 2

Thanks for the opportunity to review this narrative review of implementation strategies and outcomes of intravenous iron during and after pregnancy in LMICs. I believe that this review serves an important function by compiling results on this topic in one place, especially as multiple trials of FCM IV iron in pregnancy and postpartum along with accompanying implementation research are ongoing in LMICs. I would like to offer the following comments for the authors’ consideration.

Title – consider spelling out LMICs in the title

Line 22 – Removed “the” before postpartum.

Line 31 – If this is a narrative review, what was the role of descriptive statistics? Was this used for categorization or counting?

Line 55 and 59 – Fatigue is mentioned on both lines. Suggest combining or mentioning in only one place.

Line 67 – Women also stop taking oral iron for other reasons, including misconceptions, forgetfulness, challenges with getting adequate supply, and socioeconomic barriers.

Line 342 – 42.9% would not be considered most. The authors may want to consider rewording this.

Lines 360-361 – Please check if part of this sentence is missing.

Table 3 – This is a useful table. I have some suggestions about making it easier to read. In the types of implementation strategies column, the authors could consider making a heading for each category, especially those that repeat (e.g., Assess for readiness and identify barriers and facilitators), instead listing it several times. In the key findings column, consider using bullet points within each row (e.g., separate bullets for barriers and facilitators).

Table 5 – Similar to my comment on Table 3, in the types of implementation outcomes column, the authors could consider having heading for each outcome (e.g., acceptability, fidelity, feasibility, cost) instead of repeating in each row. The second to last row of the table has the name of an author in the types of implementation outcomes column. For the fidelity (46) row, it’s hard to understand the results in the key findings column because we don’t know what the expected duration of infusion or number of days between doses was. Cost (56) – IV iron sucrose as bolus compared to what (e.g., how many doses for the IV infusion)?

Line 428 – What does SHPs stand for?

Discussion – Consider shortening the discussion. It is quite long and could be trimmed without losing meaning.

Line 490 – Would the authors consider different countries as different settings? If so, then there is some variability in the setting.

Figures 2-4 – It’s nice to have some figures, but the information across these three figures could be shared more succinctly in a single table. Also, I noticed in Figure 2 that this refers to articles rather than studies. In some cases, I believe that some of the articles from a country are part of the same study. Would it make sense throughout the paper to refer to the number of studies rather than the number of articles?

Reviewers' comments:

Reviewer's Responses to Questions

**Comments to the Author**

1. Does this manuscript meet PLOS Global Public Health’s publication criteria?

Reviewer #1: Partly

Reviewer #2: Yes

2. Has the statistical analysis been performed appropriately and rigorously?

Reviewer #1: No

Reviewer #2: Yes

3. Have the authors made all data underlying the findings in their manuscript fully available (please refer to the Data Availability Statement at the start of the manuscript PDF file)?

Reviewer #1: Yes

Reviewer #2: Yes

4. Is the manuscript presented in an intelligible fashion and written in standard English?

Reviewer #1: Yes

Reviewer #2: Yes

Reviewer #1: 1. The search strategy to begin with itself is very incomplete where the important terms to check implementation strategies or outcomes were not looked for. This has resulted in a very vague identification of studies which were designed to look for clinical outcomes than implementation outcomes. How can formative study describing barriers and facilitators be considered as implementation strategies...these would provide basis for developing implementation strategies which are completely not available nor described. The authors understanding on implementation research sounds quite good.However they have missed the very essence in their search strategy.

2. There is no information about baseline situations in the studies in terms of availability of IV Iron type

3. The strategies described are inadequate to make any meaningful conclusions for eg. change in infrastructure - what change was done, at what level? whether this change was specific to providing IV Iron ?

4. No mention of the timelines needed for implementation

5. There is a mention of cost of IV iron as a major barrier...which is known. What implementation strategies were undertaken to address this?

6. This entire exercise needs to be repeated to clearly justify the review based on the objectives the authors set out to achieve

Reviewer #2: Thanks for the opportunity to review this narrative review of implementation strategies and outcomes of intravenous iron during and after pregnancy in LMICs. I believe that this review serves an important function by compiling results on this topic in one place, especially as multiple trials of FCM IV iron in pregnancy and postpartum along with accompanying implementation research are ongoing in LMICs. I would like to offer the following comments for the authors’ consideration.

Title – consider spelling out LMICs in the title

Line 22 – Removed “the” before postpartum.

Line 31 – If this is a narrative review, what was the role of descriptive statistics? Was this used for categorization or counting?

Line 55 and 59 – Fatigue is mentioned on both lines. Suggest combining or mentioning in only one place.

Line 67 – Women also stop taking oral iron for other reasons, including misconceptions, forgetfulness, challenges with getting adequate supply, and socioeconomic barriers.

Line 342 – 42.9% would not be considered most. The authors may want to consider rewording this.

Lines 360-361 – Please check if part of this sentence is missing.

Table 3 – This is a useful table. I have some suggestions about making it easier to read. In the types of implementation strategies column, the authors could consider making a heading for each category, especially those that repeat (e.g., Assess for readiness and identify barriers and facilitators), instead listing it several times. In the key findings column, consider using bullet points within each row (e.g., separate bullets for barriers and facilitators).

Table 5 – Similar to my comment on Table 3, in the types of implementation outcomes column, the authors could consider having heading for each outcome (e.g., acceptability, fidelity, feasibility, cost) instead of repeating in each row. The second to last row of the table has the name of an author in the types of implementation outcomes column. For the fidelity (46) row, it’s hard to understand the results in the key findings column because we don’t know what the expected duration of infusion or number of days between doses was. Cost (56) – IV iron sucrose as bolus compared to what (e.g., how many doses for the IV infusion)?

Line 428 – What does SHPs stand for?

Discussion – Consider shortening the discussion. It is quite long and could be trimmed without losing meaning.

Line 490 – Would the authors consider different countries as different settings? If so, then there is some variability in the setting.

Figures 2-4 – It’s nice to have some figures, but the information across these three figures could be shared more succinctly in a single table. Also, I noticed in Figure 2 that this refers to articles rather than studies. In some cases, I believe that some of the articles from a country are part of the same study. Would it make sense throughout the paper to refer to the number of studies rather than the number of articles?

**Do you want your identity to be public for this peer review?** For information about this choice, including consent withdrawal, please see our Privacy Policy

Reviewer #1: No

Reviewer #2: No

---

## [Decision Letter · Decision Letter 1]

20 Nov 2025

PGPH-D-25-01513R1

Implementation strategies and outcomes of intravenous iron use for treatment of anaemia during and after pregnancy in low- and middle-income countries: a scoping review

Dear Dr. Balogun,

Thank you for submitting your manuscript to PLOS Global Public Health. After careful consideration, we feel that it has merit but does not fully meet PLOS Global Public Health’s publication criteria as it currently stands. Therefore, we invite you to submit a revised version of the manuscript that addresses the points raised during the review process.

We look forward to receiving your revised manuscript.

Kind regards,

Rahul Gajbhiye, MBBS PhD

Academic Editor

Journal Requirements:

Additional Editor Comments (if provided):

Reviewers comments :

This manuscript presents a scoping review focusing on implementation strategies and outcomes of intravenous iron use in the treatment of anaemia during and after pregnancy in low- and middle-income countries (LMICs). The current burden of anaemia and the efficacy of IV iron use are evident in the treatment of anaemia. The authors followed JBI guidelines for this review and covered a broad range of databases, to provide a good overview of implementation strategy and outcomes.

1. The introduction is elaborate and descriptive, covering the well-known epidemiological facts. However, implementation gaps/data gaps need to be addressed. Authors can summarize the epidemiological background, emphasizing the inadequacy of IV iron use, research gaps in implementation, and the relevance of this scoping review to the issue.

2. Although the authors acknowledge the lack of implementation research, authors can compare the implementation differences between HICs and LMICs and the urgency of evidence from LMICs.

3. Methodology needs to be clear. Authors must clearly specify all search deviations, including changes in search terms, modifications to inclusion criteria, and any changes in analysis.

4. Include a paragraph stating the impact of differences in the type of studies, like RCTs and registry-based observational studies, on the interpretation of implementation outcomes.

5. The review reports the use of the implementation science frameworks and strategies. However, the discussion lacks adequate coverage of systemic causes that impact implementation, such as funding obstacles, training deficiencies, and policy limitations. The structural factors that limit the implementation of IV iron use in LMICs need to be elaborated upon or discussed.

6. The manuscript could be strengthened by a more profound thematic or conceptual synthesis, such as mapping barriers to specific categories, identifying regional patterns, or examining health system settings. Can add a graph/flow-chart/visual representation that summarizes barriers/facilitators/outcomes and gaps in research.

7. Additional discussion on the use of IV iron in the postpartum period would strengthen the manuscript, especially since it is highlighted in the title. The postpartum period appears to be under-represented in the included studies, likely due to the limited evidence available in the current literature. This gap should be explicitly acknowledged, along with a brief discussion of its implications for research and policy.”

8. Numbers and Tables should be made more readable. Please give clear captions to some figures. Table 3 is congested and difficult to read.

9. Regarding the PRISMA Flow Diagram, please elaborate further on the pre-2023 vs. updated 2025 search and ensure that all numbers are balanced in both the text and Figure 1.

10. Most studies are reported from India. Studies from other LMICs reflect minimal research. Please add relevant studies from other LMICs.

11. Some minor typesetting mistakes need to be corrected. Use uniform terms to refer to IV iron.

12. There are some inconsistencies in the use of abbreviations (e.g., SHP, FCM).

Reviewers' comments:

Reviewer's Responses to Questions

**Comments to the Author**

Reviewer #3: All comments have been addressed

publication criteria?

Reviewer #3: Yes

3. Has the statistical analysis been performed appropriately and rigorously?

Reviewer #3: Yes

4. Have the authors made all data underlying the findings in their manuscript fully available (please refer to the Data Availability Statement at the start of the manuscript PDF file)?

Reviewer #3: Yes

5. Is the manuscript presented in an intelligible fashion and written in standard English?

Reviewer #3: Yes

Reviewer #3: This manuscript presents a scoping review focusing on implementation strategies and outcomes of intravenous iron use in the treatment of anaemia during and after pregnancy in low- and middle-income countries (LMICs). The current burden of anaemia and the efficacy of IV iron use are evident in the treatment of anaemia. The authors followed JBI guidelines for this review and covered a broad range of databases, to provide a good overview of implementation strategy and outcomes.

1. The introduction is elaborate and descriptive, covering the well-known epidemiological facts. However, implementation gaps/data gaps need to be addressed. Authors can summarize the epidemiological background, emphasizing the inadequacy of IV iron use, research gaps in implementation, and the relevance of this scoping review to the issue.

2. Although the authors acknowledge the lack of implementation research, authors can compare the implementation differences between HICs and LMICs and the urgency of evidence from LMICs.

3. Methodology needs to be clear. Authors must clearly specify all search deviations, including changes in search terms, modifications to inclusion criteria, and any changes in analysis.

4. Include a paragraph stating the impact of differences in the type of studies, like RCTs and registry-based observational studies, on the interpretation of implementation outcomes.

5. The review reports the use of the implementation science frameworks and strategies. However, the discussion lacks adequate coverage of systemic causes that impact implementation, such as funding obstacles, training deficiencies, and policy limitations. The structural factors that limit the implementation of IV iron use in LMICs need to be elaborated upon or discussed.

6. The manuscript could be strengthened by a more profound thematic or conceptual synthesis, such as mapping barriers to specific categories, identifying regional patterns, or examining health system settings. Can add a graph/flow-chart/visual representation that summarizes barriers/facilitators/outcomes and gaps in research.

7. Additional discussion on the use of IV iron in the postpartum period would strengthen the manuscript, especially since it is highlighted in the title. The postpartum period appears to be under-represented in the included studies, likely due to the limited evidence available in the current literature. This gap should be explicitly acknowledged, along with a brief discussion of its implications for research and policy.”

8. Numbers and Tables should be made more readable. Please give clear captions to some figures. Table 3 is congested and difficult to read.

9. Regarding the PRISMA Flow Diagram, please elaborate further on the pre-2023 vs. updated 2025 search and ensure that all numbers are balanced in both the text and Figure 1.

10. Most studies are reported from India. Studies from other LMICs reflect minimal research. Please add relevant studies from other LMICs.

11. Some minor typesetting mistakes need to be corrected. Use uniform terms to refer to IV iron.

12. There are some inconsistencies in the use of abbreviations (e.g., SHP, FCM).

**Do you want your identity to be public for this peer review?** For information about this choice, including consent withdrawal, please see our Privacy Policy

Reviewer #3: **Yes: ** Dr. Abhay Gaidhane

---

## [Editor Report · Decision Letter 2]

17 Dec 2025

Implementation strategies and outcomes of intravenous iron use for treatment of anaemia during and after pregnancy in low- and middle-income countries: a scoping review

PGPH-D-25-01513R2

Dear Prof. Balogun,

We are pleased to inform you that your manuscript 'Implementation strategies and outcomes of intravenous iron use for treatment of anaemia during and after pregnancy in low- and middle-income countries: a scoping review' has been provisionally accepted for publication in PLOS Global Public Health.

Best regards,

Rahul Gajbhiye, MBBS PhD

Academic Editor